# Intersexual Differences in Behaviour and Resource Use of Specialist *Phengaris teleius* Butterflies

**DOI:** 10.3390/insects13030262

**Published:** 2022-03-06

**Authors:** Miloš Popović, Ana Golubović, Piotr Nowicki

**Affiliations:** 1Institute of Environmental Sciences, Jagiellonian University, Gronostajowa 7, 30-387 Kraków, Poland; piotr.nowicki@uj.edu.pl; 2Department of Biology and Ecology, Faculty of Sciences and Mathematics, University of Niš, Višegradska 33, 18000 Niš, Serbia; 3Institute of Zoology, Faculty of Biology, University of Belgrade, Studentski trg 16, 11000 Belgrade, Serbia; golubovic.ana@bio.bg.ac.rs

**Keywords:** flight behaviour, information processing hypothesis, mate searching, *Phengaris* (= *Maculinea*) *teleius*, resource use

## Abstract

**Simple Summary:**

In this study, we compared intersexual differences in behaviour and resource use of well-known Scarce Large Blue butterflies. The results indicated that females take short, but exceptionally regular flights, exploring available resources for oviposition. Males took longer, less regular flights, characterised by frequent interactions in search for receptive females. Interestingly, interactions between conspecific butterflies of the investigated species lasted significantly longer than with other species, especially when different sexes were involved. This suggests that individuals were able to quickly and efficiently recognise potential mating partners. Finally, the focal species showed strong association with its larval host plant while resting and nectaring, and less so with other plants with similarly coloured and shaped flowers. Females were particularly strongly restricted to the larval host plant, which provides some evidence that their sensory system shows neural limitation towards this valuable resource.

**Abstract:**

Although the behaviour of the Large Blue butterflies of the genus *Phengaris* (= *Maculinea*) is relatively well studied, most empirical data come from investigations of their dispersal and oviposition. Here, we assessed overall intersexual differences in resource use by tracking *Phengaris teleius* (Bergsträsser, 1779) individuals and recording the duration of their behaviours. Females were characterised by frequent, short flights, and devoted more time to resting and oviposition. Males engaged in numerous, but usually short interactions, and spent most of the time in flight exploring their surroundings for receptive females. Their average flight time was significantly longer compared to females. Average feeding time did not differ between the sexes but was shorter when butterflies were feeding on *Sanguisorba officinalis* L. Intraspecific interactions within *P. teleius* were three times longer than those with other insect species, and interactions between sexes were particularly long lasting. Significantly shorter interspecific interactions imply that butterflies can easily recognise conspecifics and differentiate between sexes, which offers obvious fitness benefits. Both sexes, but especially females, showed strong association with their larval host plant, and less so with other flowers of similar colour and shape. Females predominantly used their host plant for feeding, which possibly indicates neuro-sensory constraints towards this resource.

## 1. Introduction

Being monophagous or oligophagous regarding larval host plants and obligate myrmecophiles exploiting ants from the genus *Myrmica* during larval development [1,2], *Phengaris* butterflies are considered extreme specialists. Due to such a strict specialisation, these butterflies are threatened by habitat changes and over the years have become a recognisable symbol of insect conservation in Europe [3,4]. Consequently, these are also one of the most studied butterfly taxa in the world offering some very interesting behavioural facts. Namely, the association between caterpillars and ants is a prime example, where a fine arsenal of morphological, physiological and behavioural adaptations of caterpillars evolved to trick the ants into adopting them as their own. They mimic not only the appearance and movement [5], but also the smell [6] and sound [7] of the ant larvae. Adult *Phengaris* butterflies were studied mostly in the light of female behaviour leading to oviposition [8,9], or the flight behaviour that leads to dispersal in fragmented habitats [10,11,12].

Specialisation could also affect the feeding habits of adult butterflies, with *Phengaris nausithous* (Bergsträsser, 1779) and *P. teleius* being good examples. The two species are found in wetland habitats of central Europe and temperate Asia, and are regarded as sedentary, living in small, isolated populations with little exchange between local populations and with exceptionally small home ranges [12,13,14,15]. Both species oviposit on *S. officinalis* (Rosaceae), although the former prefers old inflorescences and lays multiple eggs, while the latter uses young inflorescences and usually lays a single egg at a time [8]. However, their nectaring habits are quite different. *P. nausithous* obtains nectar almost exclusively from *S. officinalis*, while *P. teleius* is known to also feed on other plants such as *Vicia craca* L. (Fabaceae) and *Serratula tinctoria* L. (Asteraceae) [16,17,18]. It is interesting to note that *P. teleius* can exploit more species of ants; thus, we can consider it less specialised compared to its closest relative [2,19,20].

*Phengaris* butterflies could also be a convenient model to explore how specialists respond to visual cues in the habitat. The information processing hypothesis states that, while generalists must process a wider range of stimuli, specialists could focus their sensory system to a single resource, spend less time choosing among resources of different quality and, consequently, face lower predation risk or make more optimal resource choices for their offspring [21]. Oviposition in butterflies is a good example in this respect. It has been demonstrated that specialist females spend less time evaluating their host plant and are reportedly able to choose host plants of better quality [22,23,24]. Females, especially those strictly specialised to a single larval host plant, could display strong preference for feeding on the same plant [25], but this might not be true for males.

We studied *P. teleius* butterflies with a special emphasis on intersexual differences in resource utilisation and behavioural patterns. While previous studies focused on dispersal behaviour and oviposition preferences of females, here we attempted to get a thorough overview of behavioural patterns, with emphasis on the duration of different activities, and we aimed to list all exploited plant resources. We expected males to spend most of the time flying in search for less mobile females. We also expected females to be more restricted to their host plant and to choose plants of similar colour and shape for feeding and resting.

## 2. Materials and Methods

To provide information on the duration of a spectrum of butterfly behaviours, two observers tracked butterflies and recorded their actions using sound recorders. The survey was conducted during four fine weather days of 8–14 August 2016, in the peak of *P. teleius’* flight period (Appendix A). Each individual was tracked for five minutes, although some butterflies were lost before the observation ended. Five behaviour categories were defined: feeding, oviposition, resting, flight and interaction. The feeding behaviour was recorded if a butterfly was probing a flower with the proboscis. Oviposition was noted if a butterfly positioned its abdomen on a host plant, but we did not distinguish successful and unsuccessful oviposition events. When a butterfly spent time on a plant without proboscis or abdomen movements, it was considered resting. Any contact of a tracked individual with other insects was considered as an interaction. For each interaction we tried to record both species names and their sex. When a butterfly was ovipositing, resting or feeding, we also recorded the plant species used. For each behaviour, we measured its duration with the precision of one second. If a butterfly flew away from the observer prior to the five-minute period, it was additionally noted. If the individual finished the entire observation period, the butterfly was then captured and marked using a permanent pen to avoid duplication. We excluded the first and the last behaviour of each individual from the analyses, since the behaviour first recorded might have started before our observation began, while the last behaviour might have continued after the five-minute observation period elapsed.

Two localities in northern Serbia were included in the study: Ludaš Lake (46.1034° N; 19.8015° E) and Selevenj Sands (46.1384° N; 19.9070° E). Ludaš (with 7 ha of suitable meadows) provides a uniform habitat, with regularly mown wet meadows surrounded by agricultural fields. Selevenj is larger (41 ha of wet meadows), with diverse, more natural grassland habitats arranged in a mosaic landscape [15]. We sampled 9–24 individuals per locality and sex. Altogether, we collected data from 77 individuals, adding up to 971 behavioural events.

In addition, we recorded individual behaviours of *P. teleius* (within the same five behavioural categories) upon capture during an intensive mark–release–recapture study in 2014 at the localities of Selevenj Sand, Ludaš Lake and Subotica Sand [15] (Appendix A). The plants used as a resource by the butterflies were identified to the species level when possible or assigned to a more general category. Although the mark–release–recapture dataset provides a broad overview of plants used in the wider region, it was not always possible to record butterfly behaviour since a large portion of individuals were scared away during capture. Thus, we excluded flight from the analysis as non-informative and were able to extract behaviour data for a total of 2222 capture events.

Data were analysed with generalised linear models or generalised linear mixed models, using the *lme4* package [26] in R [27]. Post hoc analyses were performed in the *emmeans* package [28,29]. We used Poisson family models to fit interaction time, and a binomial model to compare plant species preferences. Specimen ID was adopted as a random effect in the butterfly tracking study to account for potential individual variation in behaviour. Locality (Selevenj or Ludaš) and butterfly sex (female or male) were applied as predictor variables. Model selection was based on the Akaike Information Criterion (AIC) [30].

## 3. Results

The overview of frequencies of different behaviour categories recorded during the tracking study and their overall duration is given in Figure 1. Interestingly, females were found flying more frequently than males, but they spent less total time in flight compared to males and allocated a considerable proportion of time to oviposition. On the other hand, most behavioural observations for males were classified as interactions, but they tended to be very short; males spent most of their time in flight.

### 3.1. Average Duration of Behaviour

The best supported model explaining average times that butterflies spent in flight discarded the effect of locality and showed that males spend more time in flight (Table 1 and Appendix A). The most appropriate model describing butterfly interactions also discarded the locality as a predictor and suggested that males spend less time on interactions (Table 1 and Appendix A). Interspecific interactions were rare (10% of all the interactions) and usually included other butterfly species (62% of interspecific interactions). Two additional models that incorporated interspecific and intraspecific interactions (Table 1) suggested that the latter lasted ca. three times longer (Appendix A), and among them, interactions between different sexes were again ca. three times longer (Appendix A).

The general comparison of average feeding times showed no statistically significant differences (Table 1). Nevertheless, a more refined model was fitted to include plant species groups as a potential, additional predictor. It revealed that butterflies feed shorter on *S. officinalis* in contrast to other plants from the Asteraceae family (Table 1 and Appendix A). The best model explaining butterfly resting times suggested the interaction between sex and locality as an important predictors (Table 1) but no statistical significance could be observed for this behaviour (Appendix A).

### 3.2. Plant Preferences

Overall, *P. teleius* showed strong association with *S. officinalis* for both feeding and resting during the intensive mark–release–recapture study, and this was more evident in females (Figure 2, Table 1 and Appendix A). In total, we noted 15 plant species used by males for nectaring, with females using only 10 of them (Appendix A). At the same time, butterflies utilised a total of 31 plants and other objects for resting (Appendix A). Interestingly, some individuals were observed resting on snail shells (21 males and 13 females), and after more detailed inspection we observed a female specimen probing the shell and sucking on the snail fluids for feeding. There were two clear observations where *P. teleius* females were trying to oviposit on other plants, namely flower buds of *Cichorium intybus* L. (Asteraceae) and *S. tinctoria*, but no eggs were eventually deposited. The preference for specific flower shape and colour was notably biased towards *S. officinalis*, especially in females (Figure 2). Butterflies primarily visited flowers in the red/pink/purple colour spectrum and oval to capitulum (i.e., having a round flower head) in shape, resembling *S. officinalis*. During the butterfly tracking study, we could observe similar patterns, but males apparently preferred Asteraceae plants in contrast to females that used *S. officinalis* almost exclusively (Appendix A, Figure 3).

## 4. Discussion

The most evident result of our study was the clear difference in behavioural patterns between females and males of *P. teleius*. Females were more often found flying, but their flights were brief and linked to exploring *S. officinalis* for oviposition. Males, on the other hand, spent more time flying, and frequently (but shortly) interacted with other individuals in search for receptive females. Previous studies showed that females of *P. teleius* covered larger areas in search for suitable oviposition locations, while males tended to restrict their activity to smaller home ranges [11,31,32]. This intersexual difference was also recorded in other butterfly species, with females spending less time in flight, but covering larger distances and dispersing more often [33,34,35,36]. Such behaviour enables females to spread their eggs over a larger area, thus reducing potential intraspecific competition among their offspring [37]. On the other hand, male behaviour enables them to efficiently locate receptive females around their hatching place.

Another interesting outcome of our study was that interactions with different insect species (mostly butterflies) were much shorter in duration, while interactions between males and females of the same species lasted the longest. Mating is regarded as a costly activity for both male and female butterflies [38], and thus different mechanisms evolved to enable butterflies to distinguish conspecifics of the opposite sex (i.e., mating partners) and of own sex (i.e., mating competitors) based on colours, pheromones, courtship pattern, etc. [39]. Hence, the ability to quickly recognise individuals of different species, which should be disregarded in mate searching, is likely to increase fitness in patrolling males.

Our findings provide further evidence that the specialist *P. teleius* predominantly utilises flowers that resemble its larval host plant in terms of colour (from reddish to pink and purple) and shape (round to capitulum) [18]. Yellow *Lotus corniculatus* L. (Fabaceae) flowers were widely available in the same habitat where the tracking study was conducted, and predominantly used by another lycaenid butterfly, *Polyommatus icarus* (Rottemburg, 1775), but these were strictly avoided by *P. teleius* (authors’ unpublished results). However, it must be noted that the butterfly visual system covers the ultraviolet spectrum [40,41]; thus, colour preferences observed by humans represent an oversimplification of butterfly vision.

There was a clear intersexual separation of plant resources used for feeding. Females were tightly linked to *S. officinalis* in their plant choice, while males tended to visit other plant species. Utilising the same plant for oviposition and nectaring by females could reduce their search time and neural processing effort; thus, it can potentially increase the efficiency of their choices and be an adaptive strategy [25]. Overall, a line of behavioural evidence from the information-processing hypothesis suggests the existence of neural limitations in insects, where processing visual cues could provide an advantage for the specialists limited to less variable resources [21,25,42]. Butterfly sexes could have different nutritional requirements, in order to facilitate optimal egg development in females and flight endurance in males [43,44]. Besides, the recorded intersexual difference in preference towards nectaring plants could also decrease intraspecific competition and be an adaptive behaviour.

There was no significant intersexual difference in average time spent in individual feeding or resting behaviours, although females apparently rested more frequently. Various factors could affect female resting behaviour in butterflies, such as prolonged resting after oviposition, misinterpreting unsuccessful oviposition as resting or female age [45,46,47]. Although no significant intersexual difference in feeding time was detected in our study (possibly due to the prevailing effect of nectaring plant choice), experimental evidence suggests that males of some butterfly species are able to feed more quickly [48,49]. We could only observe that butterflies fed shorter on *S. officinalis*, which was already demonstrated for females of this species [18]. Flowers of *S. officinalis* are small and frequently visited by *P. teleius*, which might quickly wear off available nectar resources, and this potentially limits the nectaring time. *S. officinalis* nectar is rich in amino acids and limited in carbohydrates [50]. Low carbohydrate content could also explain short feeding times, since sugar is an important chemical stimulus for nectaring in butterflies [51], provides energy for flight and increases longevity and fecundity [52,53,54]. High content of amino acids is also shown for some Asteraceae plants [50], and this plant family was a common nectaring source for *P. teleius*. Only recently amino acids were recognised as important nutrients for butterfly adults, although mainly in long lived species; thus, it could be interesting to explore if amino acids stimulate *Phengaris* nectaring [52,55].

## Figures and Tables

**Figure 1 insects-13-00262-f001:**
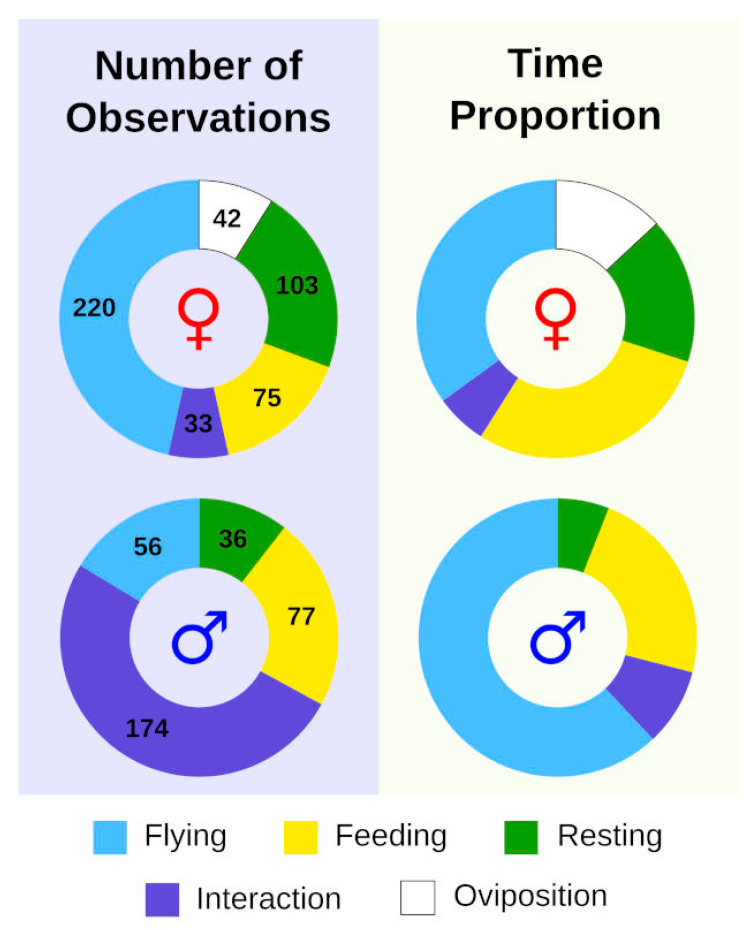
Overview of behavioural patterns of *Phengaris teleius* reflected by the numbers of observations (**left**) and duration of each type of behaviour (**right**) during the butterfly tracking study. The numbers within the graph represent sample sizes, while symbols denote females (♀) and males (♂).

**Figure 2 insects-13-00262-f002:**
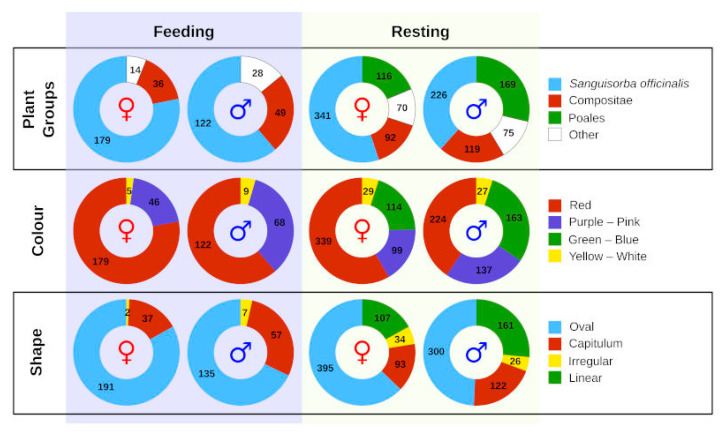
Resource use by *Phengaris teleius* butterflies for feeding and resting as recorded during a mark–release–recapture study. The plants used are classified according to species groups, flower colour and shape. The numbers within the graph represent sample sizes, while symbols denote females (♀) and males (♂).

**Figure 3 insects-13-00262-f003:**
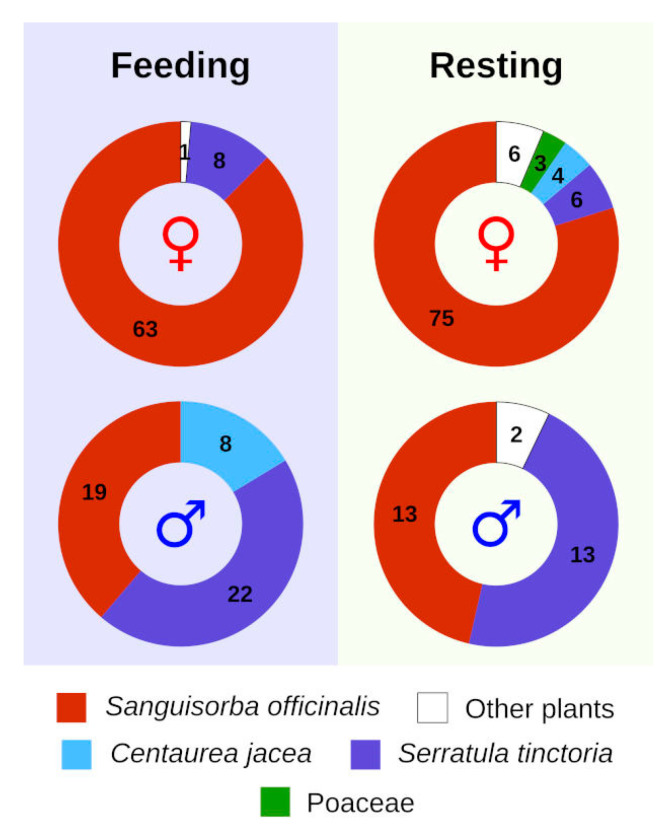
Plant species used for feeding and resting by *Phengaris teleius* during the tracking study. The numbers within the graph represent sample sizes, while symbols denote females (♀) and males (♂).

**Table 1 insects-13-00262-t001:** Summary of the Poisson models comparing average time spent on individual behaviours during the butterfly tracking study and binomial models comparing intersexual differences in plant preferences during the mark–release–recapture study.

Model Family	Model Equation
Poisson	Flying = **2.307** + **0.577** (male)
Poisson	Interaction = **2.252** − **0.780** (male)
Poisson	Interaction = 0.582 + **1.303** (intraspecific)
Poisson	Intraspecific interaction = **2.401** − **1.109** (same sex)
Poisson	Feeding = **2.860** + 0.332 (Selevenj)
Poisson	Feeding = **2.904** + **1.466** (other plants) − 0.382 (male) − 0.448 (other plants × male)
Poisson	Resting = **1.819** − 0.648 (male) − 0.129 (Selevenj) + 1.482 (male × Selevenj)
Binomial	Feeding (male) = 0.308 (Asteraceae) + **0.693** (other plants) − **0.383** (*S. officinalis*)
Binomial	Resting (male) = 0.257 (Asteraceae) + 0.069 (other plants) + **0.376** (Poales) − **0.411** (*S. officinalis*)

Numbers in the formula represent estimated coefficients, with statistically significant values (at *p* < 0.05) given in bold. Detailed information on the significant model estimates is given in Appendix A.

## Data Availability

Data are contained within Appendix A.

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
