# Peer review of "Intersexual Differences in Behaviour and Resource Use of Specialist Phengaris teleius Butterflies"

_insects, 2022, doi:10.3390/insects13030262_

Round 1
Reviewer 1 Report
This manuscript is a very strongly modified version of a draft that I had seen earlier for review. The paper now has its clear focus on just one species, the rare Phengaris teleius ant-blue butterfly, and leaves out the earlier comparsions with another species of blue butterfly (which were not that convincing). The descriptive account is concise and straight forward. It shows (not unexpectedly) that sexes differ in their time budget allocated to different behavioural activities. Presentation of the data and their evaluation is clear.
I have only a few minor suggestions where the paper should be improved.
L 25: what is meant with “on a time scale” here? I do not understand that phrase.
L 56: do not abbreviate plant genus name in the full text at first mentioning; please also give family affiliation of the plant
L 62: does that statement about width of host ant ranges still hold true in light of a more recent European synthesis paper? Please cross check with (and cite):
Tartally, A., et al. (2019). Patterns of host use by brood parasitic Maculinea butterflies across Europe. Philosophical Transactions of the Royal Society B, 374(1769), 20180202.
L 196-197: this is speculative, better omit.
L 200: typo, should read Lotus (not Lothus).
L 223: is anything known about the amount and quality of floral nectar secreted by Sanguisorba officinalis? This would be important in understanding why nectaring visits to flowers of that plant tended to be short.
See for example:
Venjakob, C., et al. (2022). Variation in nectar quality across 34 grassland plant species. Plant Biology, 24(1), 134-144.
According to that study, flower nectar of Sanguisorba is rather poor in carbohydrates, but quite rich in amino acids. If (short-lived) Phengaris butterflies are more in search of sugars (as fuel for flight) than amino acids (to build up more eggs or larger spermatophores), this scarcity in sugars could explain shorter visits.
Here and there in the paper I recognized minor inconsistencies in grammar and style, so a modest linguistic editorial ‘polishing’ would be helpful.
Author Response
Dear reviewer,
Thank you on suggestions for improving our manuscript. As before, we found this suggestions very valuable and tried to correct all the week points. Our responses (marked with “R:”) to the reviewers’ comments are listed below.
Kind regards,
Miloš Popović
This manuscript is a very strongly modified version of a draft that I had seen earlier for review. The paper now has its clear focus on just one species, the rare Phengaris teleius ant-blue butterfly, and leaves out the earlier comparsions with another species of blue butterfly (which were not that convincing). The descriptive account is concise and straight forward. It shows (not unexpectedly) that sexes differ in their time budget allocated to different behavioural activities. Presentation of the data and their evaluation is clear.
I have only a few minor suggestions where the paper should be improved.
R: We are glad that the corrected manuscript is now acceptable for publication. Although limiting the dataset to a single species also limited generalisation of the publication, the statistics are more reliable and the focus is more clear.
L 25: what is meant with “on a time scale” here? I do not understand that phrase.
R: Changed to “recording the duration of their behaviours”.
L 56: do not abbreviate plant genus name in the full text at first mentioning; please also give family affiliation of the plant
R: It was already mentioned in the Abstract, thus the genus is abbreviated. We added the family names for the mentioned plants.
L 62: does that statement about width of host ant ranges still hold true in light of a more recent European synthesis paper? Please cross check with (and cite):
Tartally, A., et al. (2019). Patterns of host use by brood parasitic Maculinea butterflies across Europe. Philosophical Transactions of the Royal Society B, 374(1769), 20180202.
R: Despite geographic variation in host ant exploitation, Phengaris nausithous is still limited to a few ant species, while others could exploit multiple resources. The mentioned paper is good reference and is included in the paper.
L 196-197: this is speculative, better omit.
R: Agreed! Removed from the manuscript.
L 200: typo, should read Lotus (not Lothus).
R: Corrected.
L 223: is anything known about the amount and quality of floral nectar secreted by Sanguisorba officinalis? This would be important in understanding why nectaring visits to flowers of that plant tended to be short.
See for example:
Venjakob, C., et al. (2022). Variation in nectar quality across 34 grassland plant species. Plant Biology, 24(1), 134-144.
According to that study, flower nectar of Sanguisorba is rather poor in carbohydrates, but quite rich in amino acids. If (short-lived) Phengaris butterflies are more in search of sugars (as fuel for flight) than amino acids (to build up more eggs or larger spermatophores), this scarcity in sugars could explain shorter visits.
R: Unfortunately we were not able to find more data on the quantity and quality of nectar. We are familiar with this publication, but didn’t discuss it before since amino acids are usually important for large and long lived species, thus we are not sure how low sugar/high amino acids could affect this short living species. More interestingly for us it that, high content of amino acids is found in some Asteraceae and (to the less extent) in Fabaceae, which are preferred by P. teleius. We tried to incorporate this into discussion and hope it is not too speculative.
Here and there in the paper I recognized minor inconsistencies in grammar and style, so a modest linguistic editorial ‘polishing’ would be helpful.
R: We send the manuscript to a colleague for language check.
Reviewer 2 Report
This is an excellent revision of the original manuscript, and I find the revised manuscript to be interesting, innovative and excellent in every way - I have no suggestion for further revision!
Author Response
Dear reviewer,
We are glad that you find that our revised version of the manuscript is improved in quality. Thank you for accepting it for publication.
Kind regards,
Miloš Popović
This manuscript is a resubmission of an earlier submission. The following is a list of the peer review reports and author responses from that submission.
Round 1
Reviewer 1 Report
The authors present an interesting database of field behaviour of males and females of a specialist butterfly and of males of an oligophagus butterfly (not a generalist according to their own definitions in the last sentence of the first paragraph of the Introduction). I think that the main problem of this MS is that the theoretical context is inappropriate. The specialist-generalist spectrum used by the authors is in terms of the number of plant species used as larval food plants (= oviposition plants), thus this difference refer mainly to female behaviour. Unfortunately, the authors were not able to obtain behavioural data of females of the “generalist” species, so the most interesting comparisons cannot be made. Other major problem that I see is that their observational data do not permit to asses directly the perceptual capabilities of the butterflies, thus the title and the idea that they are testing predictions from the “information-processing hypothesis” are not justified.
I think that the database is worth publishing, but the theoretical context (and, thus, most of the Introduction and part of the Discussion) should be changed. For me, it is hard to suggest a way to present the data of both species together. A comparison of the behaviour of both sexes of Phengaris teleius seems interesting, specially considering that this species is endangered.
I provide several specific comments and suggestions as notes in the pdf of the main manuscript.

Reviewer 2 Report
This paper reports on results of a brief observational study on the adult behaviour of two species of blue butterflies (Polyommatinae). The main idea is to compare a narrow specialist species with a more generalistic species. The main hypothesis is that the degree of specialism should be reflected in the ways butterflies allocate their time budget to different behaviours. The theoretical background of the paper, including cited literature, is okay. Presentation and statistical analysis of data is also acceptable, with just a few points to be amended (see below). However, I have a couple of more fundamental objections that must be tackled before the paper can be published.
First, a very important point relates to the flower resources in the habitats. For Ph. teleius adults obviously this is mostly identical with the larval host Sanguisorba in the first place. But which relevant host plants of Poly. Icarus did occur on the sites, in substantial densities? If, for example, Lotus corniculatus were the by far most prevalent available host on the sites at the time of your observations, than it is anything but surprising to see that the ethograms of the two species were rather similar to each other. Then, both species would just largely concentrate their nectaring efforts around the locally most prevalent resource. To address this, and generally to understand the behaviours, the nectar flower abundance at the sites needs to be included as a predictor variable (and not just the simple binomial character mown vs. unmown).
Second, the interactions must be partitioned more finely for analyses. Male:male interactions serve a different function than male:female interactions. Hence, it is necessary to split these interactions for analysis. This is particularly important since in one of the study species the sexes almost look alike (teleius), whereas in the other (Icarus) males are blue but females are (usually) brown). Hence, from a distance of a few meters, competing males are easily seen by icarus males, whereas in teleius from a distance both sexes appear visually likely very similar, also in terms of the visual system of the butterflies. And as noted above: these behaviours are embedded in different functions, viz. mate locating vs. intra-sexual competition for mates.
Apart from these general issues, I came across a number of minor points to be addressed during revision.
L 29: the study species are not so closely related to each other. A far more convincing design would for example compare, within Polyommatus, the specialist thersites or coridon or bellargus with the relative generalist icarus. Along this line, it would be interesting to learn about the longevity of butterflies in these habitats. Are the two species similar in that regard? If one were far more short-lived than the other, one would expect differences in time allocation just from that factor alone.
L 30: emphasize: adult resource use
L 35: emphasize larval host plant
L 53: "tool"? Rather/better: mechanism, trait?
L 63: should better read ‘predation risk’
L 82: Icarus is not a genuine ‘generalist’, the larvae are oligophagous on herbaceous species in one single plant family, and will not thrive on anything beyond. A true generalist (but: specialist for flowers as larval food) would for example be Celastrina argiolus within the Polyommatinae. Moreover, depending on local conditions (see above) perhaps on your study sites there was only one predominant hostplant available also for icarus? If so, its realized degree of specialism would not be so much different from the locally monophagous teleius…
L 101: icarus is not in northern Africa, this is Polyommatus celina, now regarded as a distinct species:
V Dincă et al. (2011). A combined genetic‐morphometric analysis unravels the complex biogeographical history of Polyommatus icarus and Polyommatus celina Common Blue butterflies. Molecular Ecology, 20(18), 3921-3935.
R Vodă et al. (2015). Why do cryptic species tend not to co-occur? A case study on two cryptic pairs of butterflies. PLoS ONE, 10(2), e0117802.
L 103: in the meantime, P. icarus in Canada has dispersed well beyond the province of Quebec within a few years. Thus, this statement about little range expansion no longer holds true.
L 106: this reference on ant associations in icarus is a bit outdated ; see for example
Fiedler, K. (2021). The ant associates of Lycaenidae butterfly caterpillars – revisited. Nota Lepidopterologica, 44, 159-174.
L 114: interactions MUST be differentiated between conspecifics and heteropecifics (see above). And also within species between male:male and male:female interactions.
See for one of your focal species:
Lundgren, L. 1977. The role of intra- and interspecific male:male interactions in Polyommatus icarus Rott. and some other blues (Lycaenidae). J. Res. Lepid. 16: 249-264.
L 140: instead of this mown vs. unmown category, one should better use the real nectar flower abundance (e.g. scored on a rank scale) as a co-variate.
L 184: I suggest placing information on all ‘inferior’ models in the Supplement. Instead, here present in detail the best model only, but also clearly document the Akaike weights of the various models. I did not find these values reported anywhere in the paper.
Lines 190-192 are obviously redundant, relative to Lines 180-182! Please clean up.
L 217: this feeding on ‘snail fluids’ likely represents a case of mud-puddling, which is common among lycaenid butterflies. I presume these were male individuals, since mud-puddling is heavily male-biased in lycaenids; see for example J Beck, et al. (1999), Oecologia, 119(1), 140-148.
L 223: better use the taxon name Asteraceae instead of the outdated ‘Compositae’ in the Figure and text. Which flowers precisely did you classify as having a ‘capitulum’?
L 296: Lotus flowers appear yellow to human observes, but are also highly UV-reflecting. Since P. Icarus butterflies have a visual system that extends into UV spectral range, this needs to be taken into account when speculating about the role of colour here. See for example studies done with exactly your focal species:
Knüttel, H., & Fiedler, K. (2001). Host-plant-derived variation in ultraviolet wing patterns influences mate selection by male butterflies. Journal of Experimental Biology, 204(14), 2447-2459. F. Burghardt et al. (2000). Flavonoid wing pigments increase attractiveness of female common blue (Polyommatus icarus) butterflies to mate-searching males. Naturwissenschaften, 87(7), 304-307. MP Sison-Mangus et al. (2008). The lycaenid butterfly Polyommatus icarus uses a duplicated blue opsin to see green. Journal of Experimental Biology, 211(3), 361-369.
L 307: I presently do not accept your conclusions on similar mate locating strategies, since thus far your data on interactions fail to differentiate between mate locating behaviours (male:female interactions) and ‘fights’ with potential competitors (male:male interactions). This needs to be re-worded after adapting your analyses.
Reviewer 3 Report
This comparison that purports to investigate the differences in resource “perception” between specialist and generalist butterflies does not live up to its ambition, because one important component is completely missing, namely that of the females of the generalist species. Moreover, the title is as such somewhat exaggerated because the comparison is made between one specialist and one generalist butterfly. But even if the title should be rephrased the lack of data from the female sex of the generalist butterfly, makes the contents of the paper fall short of its ambitions. This is so because, as the authors point out, the main way in which generalist and specialist butterflies are classified relates to the use of their host plants, which are chosen by the females. Therefore, the authors need to include the behaviour of females of the generalist species before this study is ready for publication.
Having said this, the authors are obviously great specialists on Phengaris teleius, and the paper is well written with a solid reference list that includes most relevant publications, and the results showing that there is no statistical difference between the males of the specialist and the generalist species, is interesting, as is the observation that the interactions between non-conspecifics are shorter in duration compared to intraspecific interactions. However, these observations are not enough to warrant publication at this point.